# Topical fluoride hesitancy and opposition are significantly and positively associated: A cross-sectional study

Joshua Lim[1], Adam C. Carle[2], Richard M. Carpiano[3], Donald L. Chi[4]*

1 University of Washington School of Dentistry, Seattle, Washington, United States of America, 2 James M. Anderson Center for Health Systems Excellence, Cincinnati Children's Hospital Medical Center, Cincinnati, Ohio, United States of America, 3 School of Public Policy, University of California Riverside, Riverside, California, United States of America, 4 Department of Oral Health Sciences, University of Washington School of Dentistry, Seattle, Washington, United States of America

* dchi@uw.edu

## Abstract

The goal of this study was to evaluate the associations between topical fluoride hesitancy and opposition to determine if hesitancy is a potential precursor to opposition. We administered an 85-item survey (11/2020-09/2021) to 1,135 caregivers that included the 20-item, 5-domain Fluoride Hesitancy Identification Tool (FHIT), from which we created five domain-specific scores of topical fluoride hesitancy (none/moderate/high for each domain); a score reflecting any topical fluoride hesitancy (moderate/high on any of the five domains); and a topical fluoride hesitancy severity score (total number of moderate/high responses to the five domains; range 0–5). The survey measured degree of topical fluoride opposition (0–10 with no = 0 and yes ≥ 1). We ran confounder-adjusted logistic regression models to evaluate associations between topical fluoride hesitancy scores and opposition. The analyses included 1,042 caregivers; mean age was 42.0 years (SD: 8.3), 78.7% were woman, and 58.3% were white. General hesitancy was reported by 82.9% of surveyed caregivers. Domain-specific hesitancy prevalence (moderate/high) was 81.3% for the necessity domain, 31.3% for chemicals, 19.5% for harm, 30.1% for uncertainty, and 25.2% for distrust. For severity, 14.7% of caregivers reported moderate/high hesitancy for all 5 domains, 7.7% for 4, 6.8% for 3, 9.3% for 2, and 43.9% for 1 domain. Opposition was reported by 39.1%. In the regression models, every hesitancy measure had a statistically significant ($p < 0.01$), positive association with opposition. In conclusion, topical fluoride hesitancy and opposition are positively associated but not identical. Additional work should elucidate the relationship between the two to inform strategies to address topical fluoride hesitancy and opposition.

**Data availability statement:** All relevant data are within the manuscript and its Supporting Information files.

**Funding:** This work was supported under the National Institute of Dental and Craniofacial Research (NIH/NIDCR) grant no. R01DE026741 (PI: DLC). The funding sources had no role in the study design, collection, analysis or interpretation of data; in the writing of the report; or in the decision to submit the article for publication.

**Competing interests:** The authors have declared that no competing interests exist.

## Introduction

Topical fluoride is an important part of preventive care provided during routine dental and medical visits [1] and is recommended by the U.S. Preventive Services Task Force and other professional organizations [2–4]. Fluoride prevents tooth decay by inhibiting demineralization of tooth enamel and promoting remineralization [5,6]. It is especially important for children at high risk for tooth decay, including those from low-income households, racial and ethnic minorities, and children with chronic medical conditions. In the U.S., 46% of children ages 2 to 19 years have tooth decay, making topical fluoride an important preventive intervention [7].

Despite strong evidence on the safety and effectiveness of topical fluoride [8,9], caregiver hesitancy and refusal are growing clinical and public health challenges [10,11]. Topical fluoride hesitancy is defined as "a delay in acceptance, thoughts of refusal, or refusal of topical fluoride despite availability" [12]. Being opposed is another term used to describe refusal. Caregivers may be hesitant about topical fluoride or refuse it for various reasons [13–16]. A 2018 study found that one-in-two caregivers had an inaccurate or incomplete understanding of fluoride varnish, a common modality of topical fluoride delivery in clinical settings [15]. Other studies have shown a significant association between topical fluoride refusal and immunization refusal [10,17], indicating general preventive care skepticism among some caregivers. This skepticism has become a more pressing issue as recent political changes associated with the U.S. presidential elections have suggested the possibility of a new federal push for eliminating water fluoridation [18]. Anecdotally, some U.S. dentists in clinical practice have reported an increase in the number of caregivers who are hesitant about or opposed to topical fluoride in the post-election period, which may increase further now that Robert F. Kennedy, Jr has been appointed as Secretary of the U.S. Department of Health and Human Services after being nominated by President Donald Trump and confirmed by the U.S. Congress. Previous qualitative research presented a five-domain model on reasons caregivers are hesitant about topical fluoride: thinking topical fluoride is not necessary, wanting to keep chemicals like fluoride out of their child's body, thinking fluoride is harmful, thinking there is too much uncertainty about fluoride, and feeling pressured to get topical fluoride [12].

The prevalence of caregivers' fluoride refusal and opposition is unknown because population-based studies do not exist. However, two smaller studies limited to mostly caregivers located in Washington state are informative. A 2014 pilot study estimated that 4.9% of caregivers refused topical fluoride based on a review of dental records for children in three dental clinics in the Seattle area [10]. A 2023 study reported that 32% of caregivers expressed some degree of opposition to topical fluoride [17].

Despite anecdotal evidence that the two phenomena of topical fluoride hesitancy and opposition are related, there is a dearth of empirical research assessing this relationship [19]. Evidence that hesitancy is a precursor to opposition would support efforts to target the former with the goal of preventing topical fluoride opposition and improving acceptance of fluoride among caregivers of children at high risk for tooth decay. The study goal was to test the hypothesis that topical fluoride hesitancy is positively associated with opposition.

## Materials and methods

### Study design and population

This was a secondary analysis of de-identified survey data collected from caregivers of children (hereafter, caregivers) in two waves from November 2020 to September 2021, with most caregivers recruited from the University of Washington Center for Pediatric Dentistry. We also recruited caregivers from additional university- or children's hospital-based pediatric dentistry clinics, private practice pediatric dentistry clinics, naturopathic medicine practices, and through social media posts and physical flyers placed on bulletin boards [20–22]. Participant inclusion criteria were: a caregiver of ≥ 1 child under 18 years old, age of ≥ 18 years old, and English speaking (N = 1,135). We excluded 93 caregivers who declined topical fluoride exclusively for financial reasons, resulting in a final sample of 1,042 caregivers [12]. Participating caregivers were entered into a raffle for gift cards, an Apple iPad, or Philips Sonicare toothbrushes [20]. We obtained informed consent from all participating caregivers in which continuing with the electronic survey after caregivers read an explanatory statement was considered consent. No written or verbal consent was obtained from participants. The study was approved by the University of Washington Institutional Review Board. The manuscript conforms to the STROBE guidelines [23].

### Measures

An 85-item survey was administered electronically using Research Electronic Data Capture (REDCap) [24] and included items to assess fluoride hesitancy and opposition [20]. There were four sets of study variables: general topical fluoride hesitancy, domain-specific topical fluoride hesitancy, severity of topical fluoride hesitancy, and topical fluoride opposition. The four hesitancy variables were the exposure variables and opposition was the outcome variable.

The validated 20-item FHIT was used to measure topical fluoride hesitancy [20,21,25]. FHIT items measure five topical fluoride hesitancy domains: thinking topical fluoride is unnecessary, wanting to keep fluoride out of their child's body, thinking fluoride is harmful, thinking there is too much uncertainty about fluoride, and feeling pressured to get topical fluoride. Each domain is measured by four items. Responses were coded: strongly disagree/not at all concerned = 0; disagree/slightly concerned = 1; agree/somewhat concerned = 2; and strongly agree/extremely concerned = 3. FHIT item number 5 was reverse coded to ensure that higher scores correspond to greater hesitancy. Each caregiver's domain-specific score was the mean value of responses for that domain.

For this study, general topical fluoride hesitancy was defined as a domain-specific score of ≥ 1 on any of the five domains. A categorical domain-specific topical fluoride hesitancy "severity" score was created using the domain-specific score: < 1 indicating no hesitancy, 1–2 indicating moderate hesitancy, and > 2 indicating high hesitancy for that specific domain. General topical fluoride hesitancy severity was defined as the number of domains in which a caregiver scored ≥1 (range: 0–5).

To measure topical fluoride opposition, caregivers were asked "On a scale of 0 to 10 with '0' being 'not at all opposed' and '10' being 'totally opposed,' how opposed are you to topical fluoride for your child?" Responses were converted to a binary outcome (0 = not opposed and ≥ 1 = opposed).

### Confounders

The analyses considered the following variables as potential confounders: child's gender, age (years), and health insurance type; caregiver's gender, age, race, ethnicity, education, religiosity (single item with a 4-point response scale: how important is religion in your life?), political ideology; and annual household income. The response categories for these items are listed in Table 1.

Parenting style was measured with four relevant items drawn from previous literature: 1) children are likely to grow up happy and healthy without much intervention from their parents; 2) parents should adjust their parenting style to the individual needs of their children; 3) finding the best educational opportunities for children is important as early as preschool;

**Table 1. Sociodemographic characteristics of all surveyed caregivers and by self-reported general topical fluoride hesitancy status (N = 1,042).**

| Sociodemographic characteristic | Total sample | General topical fluoride hesitancy status | | p |
|---|---|---|---|---|
| | | **Not hesitant** | **Hesitant** | |
| | **N (%)** | **N (%)** | **N (%)** | |
| **Sample** | 1042 (100) | 177 (17.1) | 860 (82.9) | |
| **Child gender** | | | | .92 |
| Boy | 518 (50.2) | 87 (49.7) | 427 (50.1) | |
| Girl | 514 (49.8) | 88 (50.3) | 425 (49.9) | |
| **Child age**, *mean±SD* | 7.7±4.4 | 7.7±4.5 | 7.7±4.4 | .80 |
| **Child health insurance type** | | | | **.005** |
| Private | 388 (37.9) | 82 (47.1) | 304 (35.9) | |
| Medicaid/public | 571 (55.7) | 89 (51.1) | 479 (56.6) | |
| No insurance | 22 (2.1) | 1 (0.6) | 21 (2.5) | |
| Other | 44 (4.3) | 2 (1.1) | 42 (5.0) | |
| **Caregiver gender** | | | | .36 |
| Man | 200 (20.7) | 30 (17.3) | 170 (21.4) | |
| Woman | 761 (78.7) | 141 (81.5) | 620 (78.1) | |
| Non-binary or third gender | 5 (0.5) | 2 (1.2) | 3 (0.4) | |
| Prefer to self-identify as other | 1 (0.1) | 0 (0.0) | 1 (0.1) | |
| **Caregiver age**, *mean±SD* | 42.0±8.3 | 42.9±8.0 | 41.8±8.4 | .86 |
| **Caregiver race** | | | | **<.001** |
| White | 543 (58.3) | 130 (73.4) | 413 (48.0) | |
| Black | 79 (8.5) | 4 (2.3) | 75 (8.7) | |
| Asian | 167 (17.9) | 17 (9.6) | 150 (17.4) | |
| Multiple/Other | 143 (15.3) | 18 (10.2) | 125 (14.5) | |
| **Caregiver ethnicity** | | | | .58 |
| Hispanic | 129 (13.5) | 21 (12.2) | 108 (13.8) | |
| Non-Hispanic | 826 (86.5) | 151 (87.8) | 675 (86.2) | |
| **Caregiver education** | | | | .07 |
| Less than high school diploma | 26 (2.7) | 1 (0.6) | 25 (3.2) | |
| High school diploma or equivalent | 124 (12.9) | 17 (9.9) | 107 (13.5) | |
| Some college or 2-year college degree | 277 (28.7) | 44 (25.6) | 233 (29.4) | |
| 4-year college degree | 240 (24.9) | 46 (26.7) | 194 (24.5) | |
| More than 4-year college degree | 297 (30.8) | 64 (37.2) | 233 (29.4) | |
| **Parenting style scale**, *mean±SD* | 0.6±0.5 | 0.5±0.5 | 0.7±0.5 | **.047** |
| **Caregiver religiosity** | | | | **.008** |
| Very important | 337 (34.6) | 46 (26.4) | 291 (36.4) | |
| Somewhat important | 241 (24.7) | 38 (21.8) | 203 (25.4) | |
| Not too important | 162 (16.6) | 34 (19.5) | 128 (16.0) | |
| Not at all important | 234 (24.0) | 56 (32.2) | 178 (22.3) | |
| **Caregiver political ideology** | | | | **<.001** |
| Very conservative | 32 (3.4) | 4 (2.3) | 28 (3.6) | |
| Conservative | 110 (11.7) | 14 (8.2) | 96 (12.4) | |
| Moderate | 377 (40.0) | 46 (26.9) | 331 (42.9) | |
| Liberal | 254 (26.9) | 57 (33.3) | 197 (25.5) | |
| Very liberal | 170 (18.0) | 50 (29.2) | 120 (15.5) | |

*(Continued)*

**Table 1.** (Continued)

| Sociodemographic characteristic | Total sample | General topical fluoride hesitancy status | | p |
| --- | --- | --- | --- | --- |
| | | Not hesitant | Hesitant | |
| | N (%) | N (%) | N (%) | |
| **Annual household income** | | | | .004** |
| <$15,000 | 62 (6.6) | 8 (4.7) | 54 (7.0) | |
| $15,000 to <$25,000 | 89 (9.5) | 11 (6.5) | 78 (10.2) | |
| $25,000 to <$50,000 | 201 (21.5) | 32 (18.9) | 169 (22.1) | |
| $50,000 to <$75,000 | 164 (17.5) | 24 (14.2) | 140 (18.3) | |
| $75,000 to <$100,000 | 105 (11.2) | 16 (9.5) | 89 (11.6) | |
| $100,000 to <$150,000 | 152 (16.3) | 31 (18.3) | 121 (15.8) | |
| ≥$150,000 | 162 (17.3) | 47 (27.8) | 115 (15.0) | |

*Note*: Boldface indicates statistical significance (*$p < 0.05$, **$p < 0.01$, ***$p < 0.001$).

and 4) if parents have the adequate time, resources, and knowledge, they can ensure their child's success [26]. Each item was scored on a 4-point scale (from 0="Strongly agree" to 3="Strongly disagree"). To identify a valid scoring method for these items, we evaluated the fit of a single factor model in Mplus [27] using the weighted least square mean and variance adjusted (WLSMV) estimator and treated the data as ordinal. A single factor model fit well (RMSEA=0.02; CFI=0.99; TLI=.99). However, the size of the loadings indicated that the first item on parent intervention was only weakly related to the underlying latent variable measured by the other three items. The first item's communality was 0.03, indicating that 3% of variance in the item responses was due to the underlying parenting variable. Thus, a mean parenting style score was created using responses to the remaining three items and the first item was dropped (Cronbach's alpha=0.60).

## Statistical analysis

Descriptive statistics for the study sample were generated as were the prevalence of general topical fluoride hesitancy, domain-specific topical fluoride hesitancy, severity of topical fluoride hesitancy, and topical fluoride opposition. Bivariate associations were used to identify confounders associated with each hesitancy measure and opposition. To reduce the potential for Type I error from multiple statistical comparisons, only variables significantly associated with both hesitancy and opposition were included in the final regression models (α=0.01). As a result, each model could potentially include a different set of confounders. To assess unadjusted and confounder-adjusted associations between opposition and hesitancy (i.e., general hesitancy, domain-specific hesitancy, and severity of hesitancy), binary logistic regression models were used and odds ratios (OR) and 95% confidence intervals (CI) were generated. Missing data were excluded in the analyses (ranging from 0.3% for the necessity domain unadjusted regression model to 17.1% for the uncertainty adjusted regression model). All analyses were conducted with IBM SPSS Statistics 28.0 (Chicago, IL).

## Results

### Participant sociodemographic characteristics

Child and caregiver demographics for the analytic sample are reported in Table 1. For child demographics, 50.2% were boys, mean age was 7.7 years (SD: 4.4), and 55.7% were insured by Medicaid. Among caregivers, most identified as woman (78.7%), white (58.3%), and non-Hispanic (86.5%), with a mean age of 42.0 years (SD: 8.3). Most had completed at least a four-year college degree (55.7%), had a more intensive parenting style with a mean parenting style score of 0.6 (SD: 0.5), skewed viewing religion as somewhat or very important (59.3%) and politically moderate (40.0%) or liberal (44.9%). About 44.8% of children lived in a household with an annual income greater than $75,000.

## Topical fluoride hesitancy and opposition prevalence

Nearly 83% of caregivers met criteria for general topical fluoride hesitancy (Table 1). For domain-specific topical fluoride hesitancy (S1 Table), the percentage reporting moderate or high hesitancy for each domain was 81.3% on the necessity domain (18.8% not hesitant, 75.7% moderately hesitant, and 5.6% highly hesitant); 31.3% on the chemicals domain (68.7% not hesitant, 24.4% moderately hesitant, and 6.9% highly hesitant); 30.1% on the uncertainty domain (69.9% not hesitant, 24.9% moderately hesitant, and 5.2% highly hesitant); 25.2% on the distrust domain (74.8% not hesitant, 22.1% moderately hesitant, and 3.2% highly hesitant); and 19.5% on the harm domain (80.5% not hesitant, 14.2% moderately hesitant, and 5.3% highly hesitant) (Fig 1).

For topical fluoride hesitancy severity, 17.6% of caregivers endorsed 0 hesitancy domains, 43.9% endorsed 1 domain, 9.3% endorsed 2 domains, 6.8% endorsed 3 domains, 7.7% endorsed 4 domains, and 14.7% endorsed all five hesitancy domains (S2 Table). For topical fluoride opposition, 39.1% of caregivers met or exceeded the threshold (Table 2).

## Bivariate statistics

The analyses to identify confounders indicated significant associations between all hesitancy measures (i.e., general hesitancy, domain-specific hesitancy, hesitancy severity) and opposition, on the one hand, and the following variables (all p-values <0.01): caregiver race and caregiver political ideology (Tables 1–3; S1 and S2 Tables).

## Regression models

Unadjusted and confounder-adjusted regression models revealed statistically significant associations between all topical fluoride hesitancy measures and opposition (Table 3). In adjusted models, caregivers expressing general hesitancy had a 10.78 times higher odds of opposing topical fluoride (95% CI:5.56, 20.90; $p < .001$). For all hesitancy domains, caregivers expressing moderate or high domain-level hesitancy (versus those without domain-specific hesitancy) were significantly more likely to oppose topical fluoride. For hesitancy severity, a one-unit increase in severity had a 2.29 times higher odds of opposing fluoride (95% CI: 2.04,2.57; $p < .001$).

## Discussion

The study aimed to empirically evaluate associations between different measures of topical fluoride hesitancy assessed in three different ways (general, domain-specific, and severity) and topical fluoride opposition. Consistent with the study

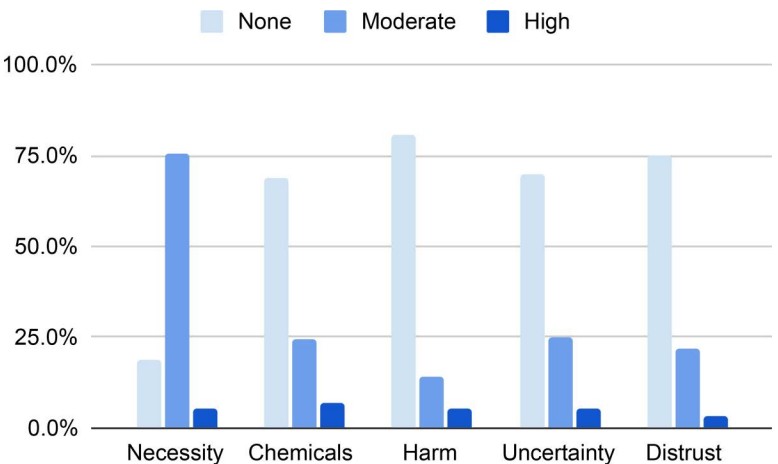

**Fig 1. Prevalence of domain-specific topical fluoride hesitancy domains in caregiver sample.**

**Table 2.  Sociodemographic characteristics of surveyed caregivers by topical fluoride opposition (N=1,042).**

| Sociodemographic characteristic | Topical fluoride opposition | | |
|---|---|---|---|
| | Not opposed | Opposed | p |
| | n (%) | n (%) | |
| **N** | 635 (60.9) | 407 (39.1) | |
| **Child gender** | | | .98 |
| Boy | 316 (50.2) | 202 (50.2) | |
| Girl | 314 (49.8) | 200 (49.8) | |
| **Child age,** *mean±SD* | 7.8±4.4 | 7.5±4.4 | .41 |
| **Child health insurance type** | | | .01* |
| Private | 260 (41.5) | 128 (32.1) | |
| Medicaid/public | 330 (52.7) | 241 (60.4) | |
| No insurance | 10 (1.6) | 12 (3.0) | |
| Other | 26 (4.2) | 18 (4.5) | |
| **Caregiver gender** | | | .63 |
| Man | 122 (20.4) | 78 (21.1) | |
| Woman | 473 (79.1) | 288 (78.0) | |
| Non-binary or third gender | 3 (0.5) | 2 (0.5) | |
| Prefer to self-identify as other | 0 (0.0) | 1 (0.3) | |
| **Caregiver age,** *mean±SD* | 42.1±8.2 | 41.9±8.6 | .16 |
| **Caregiver race** | | | <.001*** |
| White | 363 (62.5) | 180 (51.3) | |
| Black | 45 (7.7) | 34 (9.7) | |
| Asian | 83 (14.3) | 84 (23.9) | |
| Multiple/Other | 90 (15.5) | 53 (15.1) | |
| **Caregiver ethnicity** | | | .88 |
| Hispanic | 81 (13.6) | 48 (13.3) | |
| Non-Hispanic | 513 (86.4) | 313 (86.7) | |
| **Caregiver education** | | | .09 |
| Less than high school diploma | 12 (2.0) | 14 (3.8) | |
| High school diploma or equivalent | 72 (12.0) | 52 (14.2) | |
| Some college or 2-year college degree | 168 (28.1) | 109 (29.8) | |
| 4-year college degree | 145 (24.2) | 95 (26.0) | |
| More than 4-year college degree | 201 (33.6) | 96 (26.2) | |
| **Parenting style scale,** *mean±SD* | 0.6±0.5 | 0.7±0.6 | .002** |
| **Caregiver religiosity** | | | .04* |
| Very important | 194 (32.0) | 143 (38.9) | |
| Somewhat important | 145 (23.9) | 96 (26.1) | |
| Not too important | 107 (17.7) | 55 (14.9) | |
| Not at all important | 160 (26.4) | 74 (20.1) | |
| **Caregiver political ideology** | | | <.001*** |
| Very conservative | 10 (1.7) | 22 (6.3) | |
| Conservative | 61 (10.3) | 49 (14.0) | |
| Moderate | 234 (39.5) | 143 (40.7) | |
| Liberal | 182 (30.7) | 72 (20.5) | |
| Very liberal | 105 (17.7) | 65 (18.5) | |

*(Continued)*

**Table 2.** (Continued)

| Sociodemographic characteristic | Topical fluoride opposition | | |
|---|---|---|---|
| | Not opposed | Opposed | *p* |
| | n (%) | n (%) | |
| **Annual household income** | | | **.001\*\*** |
| <$15,000 | 31 (5.3) | 31 (8.9) | |
| $15,000 to <$25,000 | 47 (8.0) | 42 (12.1) | |
| $25,000 to <$50,000 | 115 (19.6) | 86 (24.8) | |
| $50,000 to <$75,000 | 107 (18.2) | 57 (16.4) | |
| $75,000 to <$100,000 | 64 (10.9) | 41 (11.8) | |
| $100,000 to <$150,000 | 106 (18.0) | 46 (13.3) | |
| ≥$150,000 | 118 (20.1) | 44 (12.7) | |

*Note*: Boldface indicates statistical significance (\*$p < 0.05$, \*\*$p < 0.01$, \*\*\*$p < 0.001$).

hypothesis, significant positive associations were observed between all the topical fluoride hesitancy measures and opposition.

First and most importantly, high degrees of topical fluoride hesitancy (82.9%) and opposition (39.1%) were observed. Although the study intent was not to assess prevalence, these estimates are informative nonetheless, as past studies have reported the prevalence of topical fluoride refusal. A 2014 study reported refusal prevalence as 12.7% and operationalized refusal with a single question asking if the caregiver had ever refused topical fluoride [10]. Both the current and previous studies were conducted in a limited number of study locations, which raises questions of how generalizable these reported prevalence rates are to other geographic and clinic sites. A 2023 study on COVID-19 vaccine hesitancy in underserved populations showed that hesitancy varied by geography [28]. Topical fluoride hesitancy and opposition may show similar geographic variation, which may be attributed in part to regional differences in political affiliation, with conservatism shown to be a determinant of vaccine hesitancy [29]. Vaccination mandates may be inconsistent with personal choice as an important value to politically conservative voters. Caregivers with conservative views may believe that public health measures, including fluoride, are an example of government overreach [30]. More caregivers in our study sample identified as liberal or very liberal (44.9%) than as conservative or very conservative (12.0%), potentially resulting in relatively lower prevalence of hesitancy in our study sample than might be observed in other more conservative geographic areas. In addition, our study sample was comprised of caregivers with a relatively high level of education – a factor that may lead to an overestimation of fluoride hesitancy and opposition. These findings underscore the importance of additional studies on fluoride hesitancy that are geographically and politically diverse as well as those that draw on caregivers from broader socioeconomic backgrounds. Cultural and psychological determinants should also be explored, given potential underlying mechanisms of these factors. Future studies are needed among caregiver populations across different states and practice settings. Similar to national studies of vaccine attitudes, a national, population-based study would help determine the prevalence of these fluoride-related phenomena, and enable assessment of potential sociodemographic and geographic variation in beliefs and behaviors [31].

Second, with regard to the association between topical fluoride hesitancy and opposition, although there is overlap between the two phenomena, topical fluoride hesitancy is not the same as topical fluoride opposition. A larger proportion of caregivers were hesitant about—but did not express any opposition to—topical fluoride. Among hesitant caregivers, 45.9% reported any opposition, whereas 97.3% of caregivers who expressed any opposition were hesitant. These findings have important clinical relevance. Large numbers of caregivers who accept topical fluoride during clinical visits for their

**Table 3. Unadjusted and confounder-adjusted odds ratios for topical fluoride opposition regressed on measures of caregiver topical fluoride hesitancy (general and domain-specific hesitancy and severity of hesitancy) (N = 1,042).**

| Hesitancy measure | Unadjusted odds ratio (95% CI) | Adjusted odds ratio[a] (95% CI) |
|---|---|---|
| **General hesitancy** | | |
| No | Referent | Referent |
| Yes | 12.82 (6.86-23.94)*** | 10.78 (5.56-20.90)*** |
| **Necessity-domain hesitancy** | | |
| None | Referent | Referent |
| Moderate | 7.52 (4.54-12.46)*** | 6.30 (3.66-10.85)*** |
| High | 37.69 (16.95-83.83)*** | 40.38 (15.86-102.84)*** |
| **Chemicals-domain hesitancy** | | |
| None | Referent | Referent |
| Moderate | 10.97 (7.82-15.38)*** | 9.90 (6.82-14.37)*** |
| High | 40.68 (17.27-95.81)*** | 52.71 (18.58-149.48)*** |
| **Harm-domain hesitancy** | | |
| None | Referent | Referent |
| Moderate | 6.12 (4.13-9.08)*** | 6.03 (3.86-9.42)*** |
| High | 10.36 (5.13-20.92)*** | 11.00 (4.98-24.28)*** |
| **Uncertainty-domain hesitancy** | | |
| None | Referent | Referent |
| Moderate | 8.70 (6.27-12.07)*** | 9.52 (6.58-13.78)*** |
| High | 15.76 (7.52-33.01)*** | 16.74 (7.22-38.80)*** |
| **Distrust-domain hesitancy** | | |
| None | Referent | Referent |
| Moderate | 8.69 (6.16-12.27)*** | 9.65 (6.54-14.24)*** |
| High | 12.26 (4.97-30.22)*** | 13.34 (4.43-40.17)*** |
| **Severity of hesitancy** | | |
| | 2.21 (1.99-2.44)*** | 2.29 (2.04-2.57)*** |

*Note*: *** *p* < .001.

[a]Confounders adjusted for each model are:

General hesitancy: child health insurance type; caregiver race and political ideology; and annual household income.

Necessity-domain hesitancy: caregiver race, parenting style, and political ideology.

Chemicals-domain hesitancy: caregiver race and political ideology; and annual household income.

Harm-domain hesitancy: caregiver race and political ideology; and annual household income.

Uncertainty-domain hesitancy: child health insurance type; caregiver race and political ideology; and annual household income.

Distrust-domain hesitancy: child health insurance type; caregiver race, parenting style, and political ideology.

Severity of hesitancy: child health insurance type; caregiver race, parenting style and political ideology.

children may have underlying concerns about fluoride that reflect a latent hesitancy. Some caregivers may not communicate these concerns to their child's health care provider, which is worrisome from a patient-provider communication perspective. This is especially important if hesitancy is a precursor to opposition. Therefore, further research is needed to assess the causal links between fluoride hesitancy and opposition, specifically via longitudinal study designs that

 

investigate the extent to which hesitancy across the various domains leads to opposition (e.g., designs that recruit new or expecting caregivers and follow them prospectively).

Third, there were substantively and statistically significant, positive associations between a variety of topical fluoride hesitancy measures and opposition. While there are no relevant studies to compare these findings to, notably, the results regarding the association of domain-specific hesitancy with opposition was generally dose-dependent. For example, the adjusted odds ratios for necessity-specific domain of hesitancy were 6.1 and 34.8 for those with moderate and high hesitancy, respectively. Longitudinal studies could help identify mechanisms underlying these associations and provide insight on the extent to which caregivers cross over from hesitancy to opposition.

Fourth, with respect to caregivers' underlying rationales for hesitancy, the findings highlight specific reasons. More than 80% of caregivers in our study believed that topical fluoride was unnecessary (the necessity domain). The next most prevalent hesitancy domain was chemicals: 31.3% of caregivers viewed fluoride as a chemical to be avoided. These findings were consistent with results from the regression models where the adjusted odds ratios were highest among these two domains (chemicals and necessity, OR = 52.71 and OR = 40.38, respectively). Furthermore, among the caregivers who met the threshold for subscribing to two domains of hesitancy (approximately 9%), all subscribed to the necessity domain and one-half of them to both the necessity and chemicals domains. These findings suggest that chairside intervention efforts should focus on clarifying topical fluoride necessity by explaining to caregivers a child's particular dental caries risk and addressing concerns caregivers may have that fluoride is a chemical. Communicating dental caries risk effectively to interested caregivers could include digital visual aids that display the teeth and show what intact but demineralized teeth look like. One possibility is the use of chairside fluorescence to show caregivers examples of early non-cavitated enamel caries lesions in their children, which are precursors to cavities that may require restorations [32]. There is also the possibility to demonstrate the beneficial effects of fluoride using these same chairside technologies. The feasibility, acceptability, and effectiveness of these approaches should be evaluated, especially with fluoride-hesitant caregivers.

Fifth, the findings provide insights to inform immediate strategies needed for caregivers of high-risk children who are hesitant about or express any opposition to topical fluoride. About 60% of caregivers in the study sample had children at high risk for caries, based on caregivers' survey responses regarding their child having a history of cavities or high likelihood of getting a cavity in the future. Given clinical recommendations that topical fluoride application is indicated for children at high risk for caries [1], hesitancy and opposition may be warranted among caregivers with children who are low-risk children [33]. However, the proportion of caregivers who expressed general hesitancy in this study (82.9%) was much higher than the proportion of caregivers with low-risk children (40%), which underscores the need to proactively address topical fluoride hesitancy in clinical settings to all caregivers. Dental caries is a multifactorial disease linked to tooth-level factors (e.g., enamel defects), intraoral environment (e.g., cariogenic bacteria, hyposalivation, xerostomia), social factors (e.g., poverty, food availability), access to preventive dental care, and the balance between two main behaviors (fluoride exposure and sugar intake) [4]. If a caregiver does not want his or her child to receive topical fluoride during a health care visit, then a clinician should discuss with the caregiver alternative forms of fluoride, including fluoride toothpaste and fluoridated water. A 2022 study reported that topical fluoride-hesitant caregivers were four times as likely to favor fluoridated toothpaste over fluoridated water, presumably because dose, frequency, and ingestion can be controlled with the former [34]. Thus, not all fluoride-hesitant caregivers are hesitant about all forms of fluoride. In fact, daily exposure to fluoridated toothpaste through toothbrushing may yield greater oral health benefits at a fraction of the cost than professionally applied topical fluoride that occurs once or twice a year. Clinical strategies to address topical fluoride hesitancy and opposition may require dentists to overcome discomfort in talking to caregivers about fluoride-related decisions. Past studies show that 42.3% of dentists reported topical fluoride refusal as a growing problem, yet 37% reported feeling uncomfortable when speaking to caregivers who refused [35]. A 2023 study presented a model that showed how dentists responded to fluoride-hesitant caregivers with varying degrees of engagement [36]. While some dentists used a patient-centered and tailored approach, others used reductionist, detached, or insistent approaches [36], which may

inadvertently increase hesitancy or lead to caregiver disengagement. Studies on evidence-based approaches to address vaccine hesitancy show that presumptive communication rather than participatory communication is associated with better vaccine uptake [37] though such approaches may not sustain behavior change in the long-term. A similar approach could be taken for fluoride, alongside other communication tools proven to be effective, such as motivational interviewing, decision aids, and diagrams for caregivers with lower health literacy [37]. On a larger scale, public health interventions such as broad and tailored community campaigns to build confidence in fluoride and positive reinforcement of fluoride uptake via immediate rewards are also evidence-based methods that may help to address hesitancy [37].

For caregivers who express any opposition to all forms of fluoride, the only feasible remaining option to reduce dental caries risk may be to address sugar intake. Sugar sweetened beverages (SSB) are the most common source of added sugars in U.S. children [38]. The American Academy of Pediatrics (AAP) recommends no SSBs for children under age 1 year and that daily 100% fruit juice intake be limited to four ounces for children ages one through three, four to six ounces for children ages four through six, and eight ounces for children ages seven through 18 [39]. A 2018 systematic review investigated education, technology, changes to physical access, and other intervention strategies to reduce SSB consumption and found that multiple strategies combined were successful, with the most common including individual education, group education, and pamphlets [40]. These interventions may be effective in educating caregivers about sugar intake. Nevertheless, while education is necessary in changing SSB intake, it is not likely to be sufficient. Behavioral interventions are needed for sustained and long-term behavior change [41]. Future research should focus on effective ways to communicate dental caries risk to caregivers as well as behavioral strategies to increase uptake of fluoride when appropriate and to reduce SSB intake.

Sixth, the analyses provide insight on modeling the relationship between hesitancy and opposition. The bivariate analyses indicate a common (e.g., caregiver race and political ideology) and a different set of confounders relevant in modeling various hesitancy measures and opposition. Future research should continue to identify relevant confounders in an effort to construct conceptually accurate epidemiologic models.

## Limitations

There are two study limitations. First, surveyed caregivers were mainly recruited from a single urban area in Washington state and most caregivers were politically liberal (leading to potential underestimation of hesitancy and opposition) but were from a higher socioeconomic background (leading to potential overestimation). As such, the direction and magnitude of any potential parameter estimation bias are unclear. However, research on topical fluoride hesitancy, opposition, and refusal is nascent and the current study addresses a critical knowledge gap in the literature while serving as a starting point for additional studies. Our sample was comparable to the latest available U.S. Census Bureau data (2020, 2023) in terms of the proportion of white individuals and income distribution [42]. Our sample had less representation from Black and Hispanic and Latino caregivers. Our sample also had notably greater representation from Asian caregivers, nearly triple the prevalence in the population (17.9% vs 6.0%) [40]. For education, the sample had less representation from individuals who had less than a high school education and greater representation from caregivers having completed college or more [42]. This provides guidance on population subgroups that might be the focus of future studies. Second, the study design was cross-sectional, limiting the ability to draw causal conclusions. However, the study operationalized multiple domains of hesitancy, providing insight for future longitudinal studies that could help disentangle cause and effect and thus advance our understanding of the mechanisms linking hesitancy and opposition.

## Conclusions

The high prevalence of fluoride hesitancy and opposition in this sample and the multifaceted nature of that hesitancy underscores the importance of dentists knowing how to identify and manage such behaviors in clinical settings. Dentists should feel empowered to talk to caregivers about fluoride-related decisions using patient-centered approaches. There

is a need for research on the efficacy of various communication strategies tailored to the specific reasons for hesitancy, which would eventually allow for the development of evidence-based clinical management strategies.

In closing, the following conclusions are offered:

1. Significant positive associations were observed between topical fluoride hesitancy and opposition, though the two are not identical.

2. A large proportion of caregivers reported some degree of hesitancy about topical fluoride (81.3%) with a smaller but still relatively large proportion reporting some degree of opposition to topical fluoride (39.1%).

3. Additional studies on clinical approaches are needed to help dentists improve fluoride-related communications, especially with caregivers of children at high risk for tooth decay who are hesitant about or express any opposition to topical fluoride.

## Supporting information

**S1 Table. Description of Sociodemographic Characteristics of Surveyed Caregivers and Domain-Specific Hesitancy with P-Values Indicating Differences in Sociodemographic Characteristics by Domain-Specific Topical Fluoride Hesitancy Category (N = 1,042).**
(DOCX)

**S2 Table. Sociodemographic Characteristics of Surveyed Caregivers and Topical Fluoride Hesitancy Severity (N = 1,042).**
(DOCX)

**STROBE Checklist. Checklist of Items That Should Be Included in Reports of Observational Studies.**
(DOCX)

## Acknowledgments

The authors would like to thank the caregivers who participated in the survey and the University of Washington Center for Pediatric Dentistry for allowing the study team to recruit participants from the clinic.

## Author contributions

**Conceptualization:** Donald L. Chi.

**Data curation:** Joshua Lim.

**Formal analysis:** Joshua Lim.

**Funding acquisition:** Donald L. Chi.

**Investigation:** Joshua Lim.

**Methodology:** Adam C. Carle, Richard M. Carpiano, Donald L. Chi.

**Software:** Adam C. Carle, Richard M. Carpiano.

**Supervision:** Donald L. Chi.

**Validation:** Adam C. Carle, Richard M. Carpiano.

**Visualization:** Joshua Lim.

**Writing – original draft:** Joshua Lim.

**Writing – review & editing:** Adam C. Carle, Richard M. Carpiano, Donald L. Chi.

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
