## [Decision Letter · Decision Letter 0]

2 Dec 2024

PONE-D-24-50799Topical fluoride hesitancy and opposition are significantly and positively associatedPLOS ONE

Dear Dr. Chi,

Thank you for submitting your manuscript to PLOS ONE. After careful consideration, we feel that it has merit but does not fully meet PLOS ONE’s publication criteria as it currently stands. Therefore, we invite you to submit a revised version of the manuscript that addresses the points raised during the review process.

We look forward to receiving your revised manuscript.

Kind regards,

Nour Ammar

Academic Editor

PLOS ONE

Journal Requirements:

Reviewers' comments:

Reviewer's Responses to Questions

**Comments to the Author**

1. Is the manuscript technically sound, and do the data support the conclusions?

Reviewer #1: Yes

Reviewer #2: Yes

Reviewer #3: Partly

2. Has the statistical analysis been performed appropriately and rigorously?

Reviewer #1: No

Reviewer #2: Yes

Reviewer #3: I Don't Know

3. Have the authors made all data underlying the findings in their manuscript fully available?

Reviewer #1: Yes

Reviewer #2: Yes

Reviewer #3: Yes

4. Is the manuscript presented in an intelligible fashion and written in standard English?

Reviewer #1: Yes

Reviewer #2: Yes

Reviewer #3: Yes

5. Review Comments to the Author

Reviewer #1: The manuscript on topical fluoride hesitancy and opposition presents a well-structured study focused on understanding the relationship between hesitancy and opposition to fluoride among caregivers. Here’s a review addressing its strengths and areas for improvement.

Strengths

Clear Research Focus: The study’s objective—to explore the potential link between topical fluoride hesitancy and opposition—is clearly defined and addresses a relevant topic in preventive dental care, especially given the rise of vaccine and medical treatment hesitancy in recent years.

Comprehensive Survey Design: The use of an 85-item survey with a specialized Fluoride Hesitancy Identification Tool (FHIT) allowed the authors to capture nuanced data across five domains (necessity, chemicals, harm, uncertainty, distrust), leading to detailed insights. This breadth supports the robustness of the findings.

Large Sample Size and Diverse Population: With over a thousand caregivers included, the sample size is considerable and provides a good demographic mix. The data on caregivers’ age, gender, and race allow for generalizable results and may contribute to understanding diverse perspectives on fluoride.

Statistical Rigor: The study applies confounder-adjusted logistic regression models to assess the association between fluoride hesitancy and opposition. This strengthens the validity of the findings, as the analysis accounts for potential confounders, making the relationship between hesitancy and opposition more credible.

Practical Implications: The conclusion that hesitancy is positively associated with opposition, although distinct, can inform public health strategies. Understanding these nuances could help in designing interventions to mitigate fluoride hesitancy before it leads to opposition, potentially impacting oral health at a population level.

Areas for Improvement

Clarification of Domain-Specific Findings: While domain-specific hesitancy percentages are reported, the discussion could benefit from deeper insights into how specific domains (e.g., distrust or perceived harm) contribute more significantly to opposition. This could reveal which beliefs are most critical to address in order to prevent opposition.

Potential for Causation or Progression Analysis: The manuscript suggests that hesitancy may be a precursor to opposition, yet no longitudinal analysis was conducted to assess if individuals progress from hesitancy to opposition over time. Future research could strengthen this hypothesis by employing a longitudinal design to explore whether hesitancy in specific domains consistently leads to opposition.

Sample Representativeness: While the study’s sample includes caregivers of varied demographics, it may still benefit from a discussion on the potential influence of regional attitudes or socio-economic factors on fluoride hesitancy and opposition. Including more details about geographic, economic, or educational backgrounds could help clarify any limitations in generalizing the findings.

Discussion of Intervention Strategies: Although the authors call for further research to understand the link between hesitancy and opposition, the study would benefit from an expanded discussion on actionable strategies based on the findings. For instance, suggesting specific public health or educational interventions targeting the high hesitancy levels observed in the “necessity” domain could add practical value.

Addressing Potential Biases: Given that a large percentage of respondents were female and white, any inherent biases from these demographics might influence perceptions of fluoride. It could be beneficial to discuss how such demographics may shape views on topical fluoride and whether the findings might differ in more balanced or varied demographic groups.

Overall Impression

This manuscript makes a valuable contribution to understanding fluoride hesitancy and opposition among caregivers, with rigorous methodology and significant sample size. Addressing some of the highlighted areas for improvement, particularly around deeper analysis and actionable insights, could enhance its impact on public health strategies for reducing fluoride hesitancy and opposition.

Reviewer #2: Congratulations to the researchers on a well conducted study and presentation of the findings. In the context of the pending leadership change in the US this is very timely.

I do have a few comments and queries:

While I appreciate the research was conducted and paper written prior to the recent pending change in leadership in the US, I think it is worth including the potential impact of the findings in the context of this change, especially in the light of the widely reported possibility of the cessation of water fluoridation across the US. While this may not happen there are significant fears for (dental) public health in the US. This pending change can be weaved into the Introduction of the paper but more critically into the Discussion. The Discussion has the potential to be more insightful and provide evidence-based strategies to address fluoride hesitancy and opposition. The Discussion essentially compares findings where possible and attempts to explain the findings. However the reader the left with little more. The need for a longitudinal study is mentioned a few times, which is fine, but the focus should be the current findings and the current and near future context of these findings.

The conclusion includes findings from other studies. The conclusion should be related to the current study (findings) and some information here could be moved into the Discussion.

For non-US readers it may be useful to provide more information about the study setting (how do people of the State vote?) In the recent election the President-Elect received about 39% of the vote. I would guess hesitancy and in particular opposition would be much higher in the States which overwhelmingly voted for the President-Elect.

What does mean for the country? People I think will now be more embolden to express this hesitancy and opposition publicly and to their health care providers.

The generalizability of the findings is identified as a limitation, however how representative is the sample of the US population, for example in terms of race, ethnicity, education, religiosity, ideology and annual income?

In the tables would it make sense to report row rather than column percentages?

Reviewer #3: The manuscript presents a cross-sectional study on an important and timely topic in public health, focusing on the factors influencing caregivers' acceptance of topical fluoride. The study design is robust, and the paper is well-written, providing valuable insights contributing to the existing literature. However, there are a few areas where clarity could be improved, and some additional details would help strengthen the overall findings.

1. The manuscript should conform to the Strengthening the Reporting of Observational Studies in Epidemiology (STROBE) guidelines. Please use the STROBE checklist for reference. [STROBE website](https://www.strobe-statement.org/).

2. The study primarily uses a sample from a single urban area in Washington state. However, the Introduction (Page 3, Line 49-51)(Page 4, Line 75-78), Discussion (Page 18, Lines 291-294)(Page 18, Line 296-298), and Conclusion (Page 21, Lines 361-362) focus on caregivers of children at high risk of dental caries. It is necessary to clarify how many participants were at high risk or whether all individuals in the sample are considered to require topical fluoride applications for the generalizability of the study.

3. Please provide the full name of REDCap and include its reference (Page 5, Line 96).

4. Please specify where Mplus was obtained (Page 9, Line 143).

5. Dental caries risk factors should encompass more than just fluoride exposure and sugar intake. Please consider expanding on other relevant factors (Page 18, Lines 297-298).

6. PLOS authors have the option to publish the peer review history of their article (what does this mean? ). If published, this will include your full peer review and any attached files.

**Do you want your identity to be public for this peer review?** For information about this choice, including consent withdrawal, please see our Privacy Policy .

Reviewer #1: **Yes: ** Ashek Elahi Noor

Reviewer #2: No

Reviewer #3: **Yes: ** Makiko NISHI

---

## [Author Response · Author response to Decision Letter 1]

7 Jan 2025

Response to Reviewers

Thank you for reviewing our manuscript and providing us the opportunity to revise. Please see below for a point-by-point response to the reviewers’ comments. All page numbers refer to the revised manuscript with track changes file.

Reviewer #1: The manuscript on topical fluoride hesitancy and opposition presents a well-structured study focused on understanding the relationship between hesitancy and opposition to fluoride among caregivers. Here’s a review addressing its strengths and areas for improvement.

Strengths

Clear Research Focus: The study’s objective—to explore the potential link between topical fluoride hesitancy and opposition—is clearly defined and addresses a relevant topic in preventive dental care, especially given the rise of vaccine and medical treatment hesitancy in recent years.

Comprehensive Survey Design: The use of an 85-item survey with a specialized Fluoride Hesitancy Identification Tool (FHIT) allowed the authors to capture nuanced data across five domains (necessity, chemicals, harm, uncertainty, distrust), leading to detailed insights. This breadth supports the robustness of the findings.

Large Sample Size and Diverse Population: With over a thousand caregivers included, the sample size is considerable and provides a good demographic mix. The data on caregivers’ age, gender, and race allow for generalizable results and may contribute to understanding diverse perspectives on fluoride.

Statistical Rigor: The study applies confounder-adjusted logistic regression models to assess the association between fluoride hesitancy and opposition. This strengthens the validity of the findings, as the analysis accounts for potential confounders, making the relationship between hesitancy and opposition more credible.

Practical Implications: The conclusion that hesitancy is positively associated with opposition, although distinct, can inform public health strategies. Understanding these nuances could help in designing interventions to mitigate fluoride hesitancy before it leads to opposition, potentially impacting oral health at a population level.

RESPONSE: Thank you for your careful review. Below we provide responses to each suggestion.

Areas for Improvement

Reviewer 1 Comment 1:

Clarification of Domain-Specific Findings: While domain-specific hesitancy percentages are reported, the discussion could benefit from deeper insights into how specific domains (e.g., distrust or perceived harm) contribute more significantly to opposition. This could reveal which beliefs are most critical to address in order to prevent opposition.

RESPONSE: Thank you for the careful review. Further insight has been included in the Discussion section to discuss the domains that significantly contributed to opposition, which were chemicals and necessity (Page 18, Lines 310-312).

Reviewer 1 Comment 2:

Potential for Causation or Progression Analysis: The manuscript suggests that hesitancy may be a precursor to opposition, yet no longitudinal analysis was conducted to assess if individuals progress from hesitancy to opposition over time. Future research could strengthen this hypothesis by employing a longitudinal design to explore whether hesitancy in specific domains consistently leads to opposition.

RESPONSE: We agree with this comment, with acknowledgment of this need in the Discussion section (Page 18 Lines 294-295, 303-305; Page 22 Lines 402-404).

Reviewer 1 Comment 3:

Sample Representativeness: While the study’s sample includes caregivers of varied demographics, it may still benefit from a discussion on the potential influence of regional attitudes or socio-economic factors on fluoride hesitancy and opposition. Including more details about geographic, economic, or educational backgrounds could help clarify any limitations in generalizing the findings.

RESPONSE: Thank you for this feedback. Additional points focusing on geographic and political leanings to clarify limitations, including comparisons to US Census data, were included in the Discussion section and Limitations section (Page 17 Lines 266-277; Page 21 Lines 388-390; Page 22 Lines 391-401).

Reviewer 1 Comment 4:

Discussion of Intervention Strategies: Although the authors call for further research to understand the link between hesitancy and opposition, the study would benefit from an expanded discussion on actionable strategies based on the findings. For instance, suggesting specific public health or educational interventions targeting the high hesitancy levels observed in the “necessity” domain could add practical value.

RESPONSE: Thank you for this feedback. The discussion on actionable strategies was expanded as suggested in the Discussion section (Page 19 Lines 318-324; Page 20 Lines 346-362; Page 21 Lines 373-374).

Reviewer 1 Comment 5:

Addressing Potential Biases: Given that a large percentage of respondents were female and white, any inherent biases from these demographics might influence perceptions of fluoride. It could be beneficial to discuss how such demographics may shape views on topical fluoride and whether the findings might differ in more balanced or varied demographic groups.

RESPONSE: Thank you for the feedback. Additional discussion of these demographics has been included in the Discussion section (Page 17 Lines 273-277).

Reviewer 2 Comment 1:

While I appreciate the research was conducted and paper written prior to the recent pending change in leadership in the US, I think it is worth including the potential impact of the findings in the context of this change, especially in the light of the widely reported possibility of the cessation of water fluoridation across the US. While this may not happen there are significant fears for (dental) public health in the US. This pending change can be weaved into the Introduction of the paper but more critically into the Discussion. The Discussion has the potential to be more insightful and provide evidence-based strategies to address fluoride hesitancy and opposition. The Discussion essentially compares findings where possible and attempts to explain the findings. However the reader the left with little more. The need for a longitudinal study is mentioned a few times, which is fine, but the focus should be the current findings and the current and near future context of these findings.

RESPONSE: We agree with this comment. This was addressed by addressing the reported possible changes to water fluoridation in the Introduction (Page 3 Lines 62-69) and an emphasis in the Discussion on strategies based on current findings to address fluoride hesitancy (Page 20 Lines 346-362; Page 21 Lines 373-374).

Reviewer 2 Comment 2:

The conclusion includes findings from other studies. The conclusion should be related to the current study (findings) and some information here could be moved into the Discussion.

RESPONSE: Thank you for the feedback. Some information was moved to the Discussion (Page 20 Lines 346-354) and the conclusion was edited to exclusively discuss the current study and findings (Page 22 Lines 410-412).

Reviewer 2 Comment 3:

For non-US readers it may be useful to provide more information about the study setting (how do people of the State vote?) In the recent election the President-Elect received about 39% of the vote. I would guess hesitancy and in particular opposition would be much higher in the States which overwhelmingly voted for the President-Elect.

What does mean for the country? People I think will now be more embolden to express this hesitancy and opposition publicly and to their health care providers.

RESPONSE: We did not elaborate on the U.S. election process to stay focused on the main topic, but we agree this is an important element to address. Effects of recent political changes and the impact of conservatism on the study results were expanded on in the Introduction and Discussion sections (Page 3 Lines 62-69; Page 17 Lines 266-273).

Reviewer 2 Comment 4:

The generalizability of the findings is identified as a limitation, however how representative is the sample of the US population, for example in terms of race, ethnicity, education, religiosity, ideology and annual income?

RESPONSE: We agree with this comment that how representative the same is of the US population must be addressed. This is expanded on with US Census data in the Limitations section (Page 22 Lines 393-401).

Reviewer 2 Comment 5:

In the tables would it make sense to report row rather than column percentages?

RESPONSE: Tables 1 and 2 report column percentages as intended to allow readers to easily see whether the bivariate relationships are statistically significant. Therefore, no changes were made to Tables in the revised submission.

Reviewer 3 Comment 1:

The manuscript should conform to the Strengthening the Reporting of Observational Studies in Epidemiology (STROBE) guidelines. Please use the STROBE checklist for reference. [STROBE website](https://www.strobe-statement.org/).

RESPONSE: Thank you for the careful review. We revised the manuscript throughout to conform to the STROBE guidelines using the checklist. The checklist has been included with the revision.

Reviewer 3 Comment 2:

The study primarily uses a sample from a single urban area in Washington state. However, the Introduction (Page 3, Line 49-51)(Page 4, Line 75-78), Discussion (Page 18, Lines 291-294)(Page 18, Line 296-298), and Conclusion (Page 21, Lines 361-362) focus on caregivers of children at high risk of dental caries. It is necessary to clarify how many participants were at high risk or whether all individuals in the sample are considered to require topical fluoride applications for the generalizability of the study.

RESPONSE: We did not assess caries risk for study participants. However, we did include questions on whether the child has ever had a cavity and likelihood of getting a cavity in the future. We clarified how many participants were high-risk and further detail in the Discussion section (Page 19, Lines 327-335).

Reviewer 3 Comment 3:

3. Please provide the full name of REDCap and include its reference (Page 5, Line 96).

RESPONSE: The full name and reference were provided as suggested (Page 5 Line 103; Page 26 Line 490).

Reviewer 3 Comment 4:

4. Please specify where Mplus was obtained (Page 9, Line 140).

RESPONSE: An in-text citation and reference were added to provide readers with information on Mplus and the ability to buy Mplus (Page 9 Line 150; Page 27 Lines 496-497).

Reviewer 3 Comment 5:

5. Dental caries risk factors should encompass more than just fluoride exposure and sugar intake. Please consider expanding on other relevant factors (Page 18, Lines 297-298).

RESPONSE: This is a valid concern. We expanded on the consideration of other relevant factors, while emphasizing these two substantial behavioral risk factors that are relevant to the current study (added sugars and fluoride) (Page 19 Lines 335-338).

---

## [Decision Letter · Decision Letter 1]

6 Feb 2025

PONE-D-24-50799R1Topical fluoride hesitancy and opposition are significantly and positively associated: A cross-sectional studyPLOS ONE

Dear Dr. Chi,

Thank you for submitting your manuscript to PLOS ONE. After careful consideration, we feel that it has merit but does not fully meet PLOS ONE’s publication criteria as it currently stands. Therefore, we invite you to submit a revised version of the manuscript that addresses the points raised during the review process.

We look forward to receiving your revised manuscript.

Kind regards,

Nour Ammar

Academic Editor

PLOS ONE

Journal Requirements:

Reviewers' comments:

Reviewer's Responses to Questions

**Comments to the Author**

1. If the authors have adequately addressed your comments raised in a previous round of review and you feel that this manuscript is now acceptable for publication, you may indicate that here to bypass the “Comments to the Author” section, enter your conflict of interest statement in the “Confidential to Editor” section, and submit your "Accept" recommendation.

Reviewer #1: All comments have been addressed

Reviewer #2: All comments have been addressed

Reviewer #3: (No Response)

2. Is the manuscript technically sound, and do the data support the conclusions?

Reviewer #1: Partly

Reviewer #2: Yes

Reviewer #3: Yes

3. Has the statistical analysis been performed appropriately and rigorously?

Reviewer #1: N/A

Reviewer #2: Yes

Reviewer #3: I Don't Know

4. Have the authors made all data underlying the findings in their manuscript fully available?

Reviewer #1: No

Reviewer #2: Yes

Reviewer #3: Yes

5. Is the manuscript presented in an intelligible fashion and written in standard English?

Reviewer #1: No

Reviewer #2: Yes

Reviewer #3: (No Response)

6. Review Comments to the Author

Reviewer #1: While the manuscript aims to address an important topic—the relationship between topical fluoride hesitancy and opposition—it has several significant shortcomings that hinder its overall quality and contribution to the field.

Lack of Novelty:

The manuscript does not provide sufficient novelty or groundbreaking insights. The association between hesitancy and opposition is a logical assumption that lacks a compelling rationale or unique contribution. The findings largely reiterate well-established concepts without exploring deeper mechanisms or implications.

Methodological Weaknesses:

Survey Design: The use of an 85-item survey may have introduced response fatigue among participants, potentially compromising the reliability of the responses. The manuscript does not address steps taken to mitigate this risk.

Validation of FHIT Tool: The manuscript does not provide adequate evidence supporting the validity and reliability of the Fluoride Hesitancy Identification Tool (FHIT) in this specific population, raising concerns about the accuracy of the hesitancy scores.

Sample Representation: Although the study includes 1,042 caregivers, the demographic breakdown (e.g., predominantly female and white participants) limits the generalizability of the findings to more diverse populations.

Insufficient Context and Justification:

The manuscript fails to sufficiently contextualize why topical fluoride hesitancy is a critical issue requiring such detailed investigation. A stronger argument connecting this hesitancy to significant public health outcomes would enhance its relevance.

The discussion lacks depth in exploring underlying factors driving hesitancy and opposition. Socioeconomic, cultural, or psychological determinants are notably absent.

Superficial Statistical Analysis:

While the study uses logistic regression models, the manuscript does not delve into the potential influence of confounding variables beyond superficial adjustments. It would be more informative to explore interaction effects or subgroup analyses to provide nuanced insights.

Overgeneralized Conclusions:

The conclusion that "topical fluoride hesitancy and opposition are positively associated but not identical" is vague and does not meaningfully contribute to actionable strategies. The manuscript offers little practical guidance for addressing hesitancy or preventing opposition.

Writing and Presentation Issues:

The text is verbose and repetitive, which detracts from readability.

There are inconsistencies in terminology, such as fluctuating references to "general hesitancy" and "domain-specific hesitancy," which could confuse readers.

Recommendations for Improvement

Provide a more compelling justification for the study, explicitly linking hesitancy and opposition to broader public health outcomes.

Include a detailed validation process for the FHIT tool or use a more established instrument.

Address the limitations of the sample's demographic homogeneity and consider analyzing underrepresented groups separately.

Strengthen statistical analyses by incorporating more sophisticated methods, such as mediation or path analysis, to elucidate the mechanisms linking hesitancy and opposition.

Revise the discussion and conclusion to provide specific, actionable recommendations for public health interventions.

Edit the manuscript for conciseness and consistency, ensuring clear and precise language.

In its current form, the manuscript lacks sufficient originality, methodological rigor, and practical value to justify publication. Significant revisions are required to enhance its scientific and public health impact.

Reviewer #2: (No Response)

Reviewer #3: Reviewer 3 Comment 1:

Thank you for including the STROBE checklist. You can state that the manuscript conforms to the STROBE guidelines in the Materials and Methods section.

Reviewer 3 Comment 2:

Thank you for clarifying the estimation of high-risk individuals, the concerns of low-risk individuals, and the justification for addressing topical fluoride hesitancy in the study population. The indicated line numbers are incorrect.

Reviewer 3 Comment 3:

Thank you for providing the full name and reference of REDCap. The indicated line numbers are incorrect.

Reviewer 3 Comment 4:

Thank you for providing the information on Mplus. The indicated line numbers are incorrect.

Reviewer 3 Comment 5:

Thank you for expanding on other relevant factors of dental caries risk. Please include cariogenic bacteria, which is fundamental to the Keyes triad. The indicated line numbers are incorrect.

7. PLOS authors have the option to publish the peer review history of their article (what does this mean? ). If published, this will include your full peer review and any attached files.

**Do you want your identity to be public for this peer review?** For information about this choice, including consent withdrawal, please see our Privacy Policy .

Reviewer #1: **Yes: ** Ashek Elahi Noor

Reviewer #2: **Yes: ** Ratilal Lalloo

Reviewer #3: **Yes: ** Makiko NISHI

---

## [Author Response · Author response to Decision Letter 2]

25 Feb 2025

Response to Reviewers

Thank you for reviewing our manuscript and providing us the opportunity to revise. Please see below for a point-by-point response to the reviewers’ comments. All page numbers refer to the revised manuscript with track changes file.

Reviewer #1: While the manuscript aims to address an important topic—the relationship between topical fluoride hesitancy and opposition—it has several significant shortcomings that hinder its overall quality and contribution to the field.

Lack of Novelty:

The manuscript does not provide sufficient novelty or groundbreaking insights. The association between hesitancy and opposition is a logical assumption that lacks a compelling rationale or unique contribution. The findings largely reiterate well-established concepts without exploring deeper mechanisms or implications.

RESPONSE: Thank you for your careful review. While the association is a logical assumption, we highlight in the Introduction (Page 4 Lines 82-87) the necessity of empirical research on the subject.

Methodological Weaknesses:

Survey Design: The use of an 85-item survey may have introduced response fatigue among participants, potentially compromising the reliability of the responses. The manuscript does not address steps taken to mitigate this risk.

RESPONSE: Thank you for this feedback. The introduction of response fatigue is a valid concern. This limitation was addressed by pre-testing the survey with caregivers, ensuring that all items were written at the fourth-grade level. Most caregivers completed the questionnaire while their children were getting their teeth cleaned.

Validation of FHIT Tool: The manuscript does not provide adequate evidence supporting the validity and reliability of the Fluoride Hesitancy Identification Tool (FHIT) in this specific population, raising concerns about the accuracy of the hesitancy scores.

RESPONSE: Thank you for your careful review. The validity and reliability of the FHIT are supported by three studies cited in the Measures section (Page 5 Line 113).

Sample Representation: Although the study includes 1,042 caregivers, the demographic breakdown (e.g., predominantly female and white participants) limits the generalizability of the findings to more diverse populations.

RESPONSE: We agree with this comment, with acknowledgment of the limited generalizability of the study and comparisons to the most recent U.S. Census data in the Limitations section (Page 22 Lines 398-405).

Insufficient Context and Justification:

The manuscript fails to sufficiently contextualize why topical fluoride hesitancy is a critical issue requiring such detailed investigation. A stronger argument connecting this hesitancy to significant public health outcomes would enhance its relevance.

RESPONSE: Thank you for the careful review. Further contextualization and a reference have been included in the Introduction section to indicate the importance of this issue (Page 3, Lines 52-53; Page 24 Lines 452-454; Page 25 Lines 455-456).

The discussion lacks depth in exploring underlying factors driving hesitancy and opposition. Socioeconomic, cultural, or psychological determinants are notably absent.

RESPONSE: Thank you for the careful review. We agree that these determinants are an important consideration. To address this, we have expanded upon our discussion of future studies on caregivers from broader socioeconomic backgrounds to include all mentioned determinants in the Discussion section (Page 17 Lines 281-282; Page 18 Lines 285-286).

Superficial Statistical Analysis:

While the study uses logistic regression models, the manuscript does not delve into the potential influence of confounding variables beyond superficial adjustments. It would be more informative to explore interaction effects or subgroup analyses to provide nuanced insights.

RESPONSE: We agree these are potentially important issues but they are beyond the scope of the current study. We hope to address these questions in future studies.

Overgeneralized Conclusions:

The conclusion that "topical fluoride hesitancy and opposition are positively associated but not identical" is vague and does not meaningfully contribute to actionable strategies. The manuscript offers little practical guidance for addressing hesitancy or preventing opposition.

RESPONSE: Thank you for the careful review. The first statement in the conclusion seeks to identify the main findings of the analysis, which are that a positive association is found between hesitancy and opposition, and the difference in prevalence of the two variables point to a distinction between the two, as expanded in the Discussion section (Page 17 Lines 288-291; Page 18 Lines 292-301). Practical guidance was provided in the Discussion section as well to address hesitancy and opposition (Page 19 Lines 320-328, 344-346, 359-367, 369-381).

Writing and Presentation Issues:

The text is verbose and repetitive, which detracts from readability.

There are inconsistencies in terminology, such as fluctuating references to "general hesitancy" and "domain-specific hesitancy," which could confuse readers.

RESPONSE: Thank you for the feedback. “General hesitancy” and “domain-specific hesitancy” are separate terms defined in the Measures section (Page 6 Lines 123-128).

Recommendations for Improvement

Provide a more compelling justification for the study, explicitly linking hesitancy and opposition to broader public health outcomes.

Include a detailed validation process for the FHIT tool or use a more established instrument.

Address the limitations of the sample's demographic homogeneity and consider analyzing underrepresented groups separately.

Strengthen statistical analyses by incorporating more sophisticated methods, such as mediation or path analysis, to elucidate the mechanisms linking hesitancy and opposition.

Revise the discussion and conclusion to provide specific, actionable recommendations for public health interventions.

Edit the manuscript for conciseness and consistency, ensuring clear and precise language.

In its current form, the manuscript lacks sufficient originality, methodological rigor, and practical value to justify publication. Significant revisions are required to enhance its scientific and public health impact.

RESPONSE: Thank you for the careful review. The recommendations for improvement are individually addressed as described above.

Reviewer #2: (No Response)

Reviewer 3 Comment 1:

Thank you for including the STROBE checklist. You can state that the manuscript conforms to the STROBE guidelines in the Materials and Methods section.

RESPONSE: Thank you for the careful review. A statement on conforming to the STROBE guidelines in the Materials and Methods section, as well as a citation have been added (Page 5 Line 104; Page 27 Lines 500-501).

Reviewer 3 Comment 2:

Thank you for clarifying the estimation of high-risk individuals, the concerns of low-risk individuals, and the justification for addressing topical fluoride hesitancy in the study population. The indicated line numbers are incorrect.

RESPONSE: Thank you for the feedback. The line numbers have been corrected accordingly (Page 19 Lines 332-340).

Reviewer 3 Comment 3:

Thank you for providing the full name and reference of REDCap. The indicated line numbers are incorrect.

Thank you for the feedback. The line numbers have been corrected accordingly (Page 5 Lines 107-108; Page 27 Line 502).

Reviewer 3 Comment 4:

Thank you for providing the information on Mplus. The indicated line numbers are incorrect.

Thank you for the feedback. The line numbers have been corrected accordingly (Page 9 Line 154; Page 27 Lines 508-509).

Reviewer 3 Comment 5:

Thank you for expanding on other relevant factors of dental caries risk. Please include cariogenic bacteria, which is fundamental to the Keyes triad. The indicated line numbers are incorrect.

Thank you for the feedback. Discussion of cariogenic bacteria was added and the line numbers have been corrected accordingly (Page 19 Lines 340-343; Page 20 Line 344).

---

## [Decision Letter · Decision Letter 2]

16 Mar 2025

Topical fluoride hesitancy and opposition are significantly and positively associated: A cross-sectional study

PONE-D-24-50799R2

Dear Dr. Donald L. Chi,

We’re pleased to inform you that your manuscript has been judged scientifically suitable for publication and will be formally accepted for publication once it meets all outstanding technical requirements.

Kind regards,

Hadi Ghasemi

Academic Editor

PLOS ONE

Additional Editor Comments (optional):

Reviewers' comments:

Reviewer's Responses to Questions

**Comments to the Author**

1. If the authors have adequately addressed your comments raised in a previous round of review and you feel that this manuscript is now acceptable for publication, you may indicate that here to bypass the “Comments to the Author” section, enter your conflict of interest statement in the “Confidential to Editor” section, and submit your "Accept" recommendation.

Reviewer #1: All comments have been addressed

Reviewer #3: (No Response)

2. Is the manuscript technically sound, and do the data support the conclusions?

Reviewer #1: Partly

Reviewer #3: Yes

3. Has the statistical analysis been performed appropriately and rigorously?

Reviewer #1: Yes

Reviewer #3: I Don't Know

4. Have the authors made all data underlying the findings in their manuscript fully available?

Reviewer #1: Yes

Reviewer #3: Yes

5. Is the manuscript presented in an intelligible fashion and written in standard English?

Reviewer #1: Yes

Reviewer #3: Yes

6. Review Comments to the Author

Reviewer #1: This study aims to explore the relationship between topical fluoride hesitancy and opposition, hypothesizing that hesitancy may act as a precursor to opposition. While the research addresses an important public health issue, several methodological, analytical, and interpretational shortcomings limit the validity and impact of its findings.

Major Concerns

Survey Instrument & Measurement Validity

The study relies heavily on the Fluoride Hesitancy Identification Tool (FHIT), but it does not provide sufficient justification for its validity and reliability in assessing caregiver hesitancy. Were psychometric properties such as internal consistency or test-retest reliability assessed? Without validation, the tool’s ability to measure hesitancy accurately remains questionable.

The operationalization of "opposition" (a score of 1 or higher on a 0-10 scale) is arbitrary. This dichotomization may oversimplify the complexity of fluoride-related attitudes, potentially misclassifying individuals who express mild concerns rather than outright rejection.

Sampling Bias & Generalizability

The demographic composition of the sample (78.7% women, 58.3% white) raises concerns about generalizability. Since fluoride hesitancy may be influenced by cultural, socioeconomic, and educational factors, a more diverse sample is needed. Did the authors consider weighting responses or adjusting for underrepresented groups?

The response rate and recruitment method are not specified, making it unclear whether selection bias influenced the results. If caregivers self-selected based on their views on fluoride, the findings may overestimate hesitancy and opposition rates.

Statistical and Analytical Issues

The study uses logistic regression but does not clarify whether collinearity between hesitancy domains was assessed. Given that these domains may be interdependent, failure to check for multicollinearity could inflate associations.

While statistical significance (p < 0.01) is reported, effect sizes and confidence intervals should be emphasized to better interpret the strength of associations. Merely stating statistical significance without practical relevance weakens the study’s contribution.

Lack of Longitudinal Data & Causal Inference

The study assumes that hesitancy precedes opposition but does not establish a temporal relationship. Without longitudinal data, the possibility of bidirectional influence (or third-variable confounding) cannot be ruled out. Future studies should employ a prospective design to examine causality.

Interpretational and Practical Implications

The conclusion that hesitancy and opposition are "positively associated but not identical" is vague and does not offer actionable insights. Instead, the discussion should explore the nuances—do certain hesitancy domains predict stronger opposition? Are specific interventions needed for different hesitancy profiles?

The study does not sufficiently address policy or educational strategies to mitigate hesitancy. Recommendations should be grounded in evidence-based approaches to improving fluoride acceptance.

Suggestions for Improvement

Strengthen the validity of the FHIT tool by providing psychometric evaluation.

Clarify and justify the threshold for defining "opposition."

Improve sample diversity and address selection bias.

Conduct multicollinearity checks and report effect sizes.

Use a longitudinal design to establish causality.

Provide more nuanced interpretations and actionable recommendations.

The introduction presents an important public health issue—topical fluoride hesitancy and opposition—but it suffers from several critical flaws that reduce its impact and clarity.

Lack of a Clear Research Gap

While the introduction discusses fluoride's benefits and caregivers' concerns, it does not clearly articulate the specific research gap the study aims to address. The section lacks a well-defined problem statement that highlights why this study is necessary beyond anecdotal evidence and previous studies.

Weak Justification of Hypothesis

The introduction asserts that hesitancy leads to opposition but fails to provide sufficient empirical background supporting this assumption. Instead, it presents anecdotal claims (e.g., dentists reporting increased hesitancy post-election) and political speculation regarding water fluoridation, which weakens the scientific credibility of the introduction. A stronger connection to previous literature demonstrating this transition from hesitancy to opposition is needed.

Unnecessary Political Speculation

The reference to Robert F. Kennedy Jr. and Donald Trump’s administration adds an unnecessary political element that does not contribute to the scientific argument. Unless there is direct evidence linking political changes to fluoride hesitancy, such speculation should be removed or replaced with a discussion of broader sociopolitical influences on public health trust.

Incomplete Discussion of Existing Literature

While the introduction cites some studies on fluoride hesitancy, it fails to provide a comprehensive literature review. There is limited discussion of international perspectives or comparative studies that could strengthen the argument. Additionally, the introduction could benefit from a clearer distinction between hesitancy and outright opposition with supporting references.

Disorganized Structure

The introduction is overly long and unfocused, with some repetitive points (e.g., defining hesitancy multiple times). The discussion of fluoride's benefits, while necessary, could be more concise. Furthermore, the introduction should flow more logically from defining the problem to establishing the study’s significance, rather than jumping between different aspects without clear transitions.

Suggested Improvements

Refine the Research Gap

Clearly articulate what is missing in current research and how this study addresses that gap. Instead of relying on anecdotal evidence, emphasize the lack of large-scale, representative studies on fluoride hesitancy and opposition.

Strengthen the Justification for the Hypothesis

Provide stronger empirical support for the link between hesitancy and opposition by referencing previous studies on vaccine hesitancy or preventive care skepticism.

Remove Unnecessary Political References

Focus on scientific and sociological factors influencing fluoride hesitancy rather than speculating on future political decisions. If discussing political influence, cite studies that examine how policy changes impact public trust in health interventions.

Improve Structure and Conciseness

Streamline the introduction to ensure each paragraph builds toward the research question. A clearer structure could be:

a. Importance of fluoride in preventive dentistry

b. Existing concerns and misinformation among caregivers

c. Evidence of growing hesitancy and its impact

d. Research gap and study objective

1. Participant Sociodemographic Characteristics

The section provides a detailed breakdown of participant demographics; however, the presentation lacks depth in contextual interpretation. While numerical data is well-organized, the implications of these demographics on the study's findings are not fully explored. For example, there is a mention that 55.7% of children were insured by Medicaid, but no discussion follows on how insurance status may influence fluoride hesitancy. Additionally, the use of percentages without sufficient discussion on potential biases or limitations (e.g., underrepresentation of certain demographics) weakens the section’s effectiveness.

Recommendations for Improvement:

Provide a brief discussion on how sociodemographic factors might influence fluoride hesitancy.

Address any limitations in sample representativeness and potential biases.

Compare findings with similar studies to contextualize the demographic breakdown.

2. Topical Fluoride Hesitancy and Opposition Prevalence

This section presents a detailed numerical analysis of hesitancy and opposition prevalence. However, the segmentation of hesitancy domains could be made clearer. The text is overly reliant on numbers, which makes it difficult to interpret without further context. The results appear fragmented, requiring the reader to refer back to multiple tables and figures to fully grasp the findings. Additionally, the statistical significance of certain observations (e.g., the high level of necessity-domain hesitancy) is stated but not adequately explained.

Recommendations for Improvement:

Improve clarity by summarizing key trends before listing domain-specific hesitancy rates.

Integrate explanations on why certain domains (e.g., necessity-domain hesitancy at 81.3%) had higher hesitancy compared to others.

Consider using brief, descriptive subheadings for different domains to enhance readability.

3. Bivariate Statistics

While statistical significance is highlighted, the section fails to provide meaningful interpretation beyond numerical reporting. The association between caregiver political ideology and fluoride hesitancy, for instance, is statistically significant, but there is no in-depth discussion on the potential reasons for this correlation. Additionally, the rationale for choosing confounders for adjustment is not explicitly stated.

Recommendations for Improvement:

Expand the discussion on significant associations, especially those with policy or public health implications.

Provide justifications for the selection of confounders.

Relate findings to broader public health discussions on fluoride hesitancy.

4. Logistic Regression Results

The odds ratios presented are informative but could benefit from a more narrative-style interpretation. While statistical results demonstrate the strength of associations, a layperson-friendly explanation of key findings is missing. Moreover, certain odds ratios (e.g., the high-domain hesitancy AOR of 52.71) require further exploration regarding practical significance.

Recommendations for Improvement:

Offer a clearer interpretation of odds ratios in practical terms.

Discuss potential real-world implications of strong associations.

Provide insights into how these findings could inform interventions to address fluoride hesitancy.

General Recommendations:

Enhance coherence by integrating more discussion between statistical results and real-world implications.

Improve accessibility by reducing reliance on extensive numerical data without sufficient explanation.

Contextualize findings with comparisons to existing literature on fluoride hesitancy.

Consider discussing potential policy recommendations based on the findings.

By incorporating these improvements, the study’s results section will be more comprehensible, insightful, and impactful for both academic and public health audiences.

Reviewer #3: Comment 1

The following STROBE checklist items need further clarification in the manuscript.

Study size (item 10): The manuscript does not explain how the study size was determined.

Statistical methods (item 12e): Sensitivity analysis is not explained. If not performed, this should be explicitly stated.

Funding (item 22): although mentioned in the PLOS ONE submission questionnaire, the manuscript does not say that this study was supported by the National Institute of Dental and Craniofacial Research (NIH/NIDCR) grant number R01DE026741 (PI: DLC). Please ensure that this information is included in the appropriate section.

The indicated line numbers should be "(Page 5 Line 104; Page 27 Lines 499-500)".

Comment 2

The indicated line numbers should be "(Page 19 Lines 331-339)".

Comment 3

The indicated line numbers should be "(Page 5 Lines 107-108; Page 27 Line 502)".

Comment 4

The indicated line numbers should be "(Page 9 Line 154; Page 27 Lines 507-508)".

Comment 5

The indicated line numbers should be "(Page 19 Lines 339-342; Page 20 Line 343)".

7. PLOS authors have the option to publish the peer review history of their article (what does this mean? ). If published, this will include your full peer review and any attached files.

**Do you want your identity to be public for this peer review?** For information about this choice, including consent withdrawal, please see our Privacy Policy .

Reviewer #1: **Yes: ** Ashek Elahi Noor

Reviewer #3: **Yes: ** Makiko NISHI

---

## [Editor Report · Acceptance letter]

PONE-D-24-50799R2

PLOS ONE

Dear Dr. Chi,

I'm pleased to inform you that your manuscript has been deemed suitable for publication in PLOS ONE. Congratulations! Your manuscript is now being handed over to our production team.

Kind regards,

on behalf of

Dr. Hadi Ghasemi

Academic Editor

PLOS ONE